# Influence of Transportation on Stress Response and Cellular Oxidative Stress Markers in Juvenile Meagre (*Argyrosomus regius*)

**DOI:** 10.3390/ani13203288

**Published:** 2023-10-21

**Authors:** Martina Bortoletti, Elisa Fonsatti, Federico Leva, Lisa Maccatrozzo, Cristina Ballarin, Giuseppe Radaelli, Stefano Caberlotto, Daniela Bertotto

**Affiliations:** 1Department of Comparative Biomedicine and Food Science (BCA), University of Padova, Viale dell’Università 16, I-35020 Legnaro, Padova, Italy; martina.bortoletti@unipd.it (M.B.); elisa.fonsatti@phd.unipd.it (E.F.); federico.leva88@gmail.com (F.L.); lisa.maccatrozzo@unipd.it (L.M.); cristina.ballarin@unipd.it (C.B.); daniela.bertotto@unipd.it (D.B.); 2Valle Cà Zuliani Società Agricola Srl, Via Pila 48, I-45018 Pila di Porto Tolle, Rovigo, Italy; stefano.caberlotto@vallecazuliani.it

**Keywords:** 24 h transport, fish welfare, HSP70, HNE, NT, 8-OHdG

## Abstract

**Simple Summary:**

The development of aquaculture has led to an increased attention towards the welfare of farmed fish. Transportation, a routine procedure in this industry, can be stressful for fish due to handling, air exposure, confinement, and reduced oxygen levels. Therefore, it is crucial to gain insights into optimal practices for organizing transportation, ensuring the well-being of these animals and minimizing excessive stress. The objective of this study was to assess the stress response of meagre juveniles during commercial transport from Monfalcone to Stintino (Italy), which lasted 24 h. Welfare evaluation included measuring cortisol levels in muscle and assessing the potential cellular localization of oxidative stress markers in various organs and tissues. Significant increases in cortisol were observed in the fish during transfer from housing to transport tanks and in subsequent samplings at different phases of transport. This finding suggests that either the animals had insufficient time to recover or the transport conditions were suboptimal. However, in terms of oxidative stress, overall immunohistochemical analysis did not reveal variation in marker localization before and after transport, indicating that the stress experienced by the fish during transportation might have been relatively mild.

**Abstract:**

In aquaculture, the transportation of live fish is a crucial but stress-inducing practice, necessitating a thorough understanding of its impact on fish welfare. This study aimed to assess the physiological stress response of meagre (*Argyrosomus regius*) juveniles during a 24 h commercial transport by quantifying muscle cortisol levels using a specific radioimmunoassay. Additionally, an immunohistochemical approach was used to detect and localize the cellular distribution of oxidative-stress-related biomarkers within various tissues and organs. The results demonstrated a significant increase in muscle cortisol levels following the loading procedure, remaining elevated above basal levels throughout the 24 h transport period. This effect may be attributed to either insufficient time for recovery from the loading stress or prolonged transportation-related stress. Immunostaining for all the antibodies we examined was observed in multiple tissues and organs, but we found no notable variations among the various transport phases. In conclusion, the observed stress response appears to be mainly linked to loading stress and the transport process itself, emphasizing the importance of implementing appropriate operational procedures to safeguard fish well-being during transport. Nonetheless, the unaltered distribution of oxidative stress markers between the control and transported groups suggests that the experienced stress might be within tolerable limits.

## 1. Introduction

In aquaculture, transportation is a crucial process for transferring juveniles from the hatchery to the commercial farm, where they are reared before being sent to the food market. Furthermore, in farms located far from the slaughterhouse, transportation becomes essential to convey the fish to the facility where they are sacrificed and prepared for sale [1]. Changes in water quality can adversely affect the physiology of fish [2] as well as confinement inside transport tanks, especially when fish are transported at high densities, exacerbates the situation, causing skin abrasions, loss of scales and mucus, and creating entry routes for pathogens, thereby increasing the likelihood of disease incidence [3,4].

Any stressor elicits a physiological response in animals known as the stress response [5], which unfolds in three distinct stages: primary, secondary and tertiary response. In fish, the primary response encompasses the triggering of two neuroendocrine axes. On one hand, the hypothalamus–sympathetic–chromaffin axis stimulates the release of catecholamines, primarily adrenaline and noradrenaline, from chromaffin cells. On the other hand, the hypothalamus–pituitary–interrenal axis prompts the secretion of corticosteroids, with cortisol as the primary hormone, from the interrenal tissue. In fish, as in mammals, these coordinated responses work in tandem to mobilize energy reserves, addressing the increased metabolic demands arising from the stressful situation [6].

Following the primary response, a secondary one develops, characterized by the effects produced by hormones once they reach their target tissues through the bloodstream. At the cellular level, stress frequently induces an elevation in Heat Shock Proteins’ (HSPs) synthesis [7]. HSPs play a crucial role in cellular protection and stress response mechanisms [8]. The synthesis of HSP70, a member of the HSP family, is induced in fishes under stressful conditions, such as transport-related stress [9,10].

The secondary response to stress may also lead to a decline in the antioxidant defense capacity. This decrease arises from an imbalance between reactive oxygen species (ROS), which are byproducts of normal metabolic processes, and antioxidants that typically neutralize ROS [11]. Consequently, the imbalance gives rise to oxidative stress, resulting in lipid oxidation, the formation of protein carbonyls, genetic damage, and ultimately culminating in cellular demise [12,13]. Lipid peroxides are utilized as unstable indicators of oxidative stress, undergoing decomposition to produce more complex and reactive compounds, among which 4-Hydroxy-2-nonenal (HNE) is commonly employed as a biomarker for lipid peroxidation [14]. Another noteworthy ROS is the superoxide radical, which, when combined with nitric oxide, gives rise to peroxynitrite, a potent oxidant with the capacity to impact DNA, lipids, and proteins. 3-Nitrotyrosine (NT), arising from the oxidative transformation of the amino acid tyrosine, is employed as a relatively stable marker for peroxynitrite production [15]. DNA damage induced by oxidative stress can encompass the occurrence of single- and double-strand breaks, as well as base modifications such as the oxidation of deoxyguanosine to form 8-Hydroxy-2′-deoxyguanosine (8-OHdG) [16]. 8-OHdG represents a prominent biomarker of DNA damage, and it is associated with a well-established mutagenic potential, particularly involving G to T transversions [17]. The presence of all these markers has been previously observed in various fish species exposed to diverse stressors, including transportation [18], handling, temperature fluctuations [19], ammonia elevation [20], and environmental contaminants [21,22].

If the stress response continues for an extended period, a tertiary response, or the so-called chronic stress response, takes place. This response involves comprehensive changes at the organism level, including reduced growth rates and reproductive capacities, behavioral modifications, and immunosuppression. Ultimately, the chronic stress response may even lead to the animal’s death [6].

In this study, we investigated the effects of a 24 h transport on the stress response in meagre (*Argyrosomus regius*) juveniles by using radioimmunoassay to measure muscle cortisol levels and immunohistochemistry to detect the presence and intracellular localization of HSP70, HNE, NT, and 8-OHdG in various organs and tissues. *Argyrosomus regius* has recently been introduced into aquaculture, gaining popularity due to its notably high growth rates and meat quality, leading to its widespread adoption [23]. Consequently, there is currently limited information available regarding the stress response of this species following commercial transport. To our knowledge, the sole available study on this subject originates from our research group [18]. Our findings will hopefully contribute to a deeper understanding of the challenges posed by live fish transportation in aquaculture and offer insights into potential strategies to enhance fish welfare during this critical phase.

## 2. Materials and Methods

The animals included in this study were sampled at different time intervals during a commercial transport from the hatchery to the fattening facility. All animals used in this study were handled in accordance with the guidelines of the Council Regulation (EC) No. 1/2005 which establishes transport conditions to be fulfilled in order to ensure the welfare of animals, and Directive 2010/63/EU, which regulates the use of animals for experimental and other scientific purposes.

### 2.1. Transport Conditions, Animals and Sampling

*Argyrosomus regius* juveniles were transported from Monfalcone (Gorizia, Italy) to Stintino (Sassari, Italy) following a 72 h fasting period. The fish were gently crowded using nets and then rapidly transferred by hand-nets to the tanks of the vehicle for transportation by trained personnel. The entire transportation process lasted for 24 h and was conducted using a commercial truck equipped with ten specialized tanks (2.7 m^3^), insulated and equipped with aeration and temperature control systems. The fish weighed 7.5 g, and the tank density was maintained at 20 kg/m^3^. During the transport, which took place in the summer season (mid-July), the temperature and level of oxygen dissolved in water were, respectively, 19–24 °C and 16–20 ppm.

Sampling occurred at four phases: before transport (control), after loading (which lasted 4 h), during the travel (16 h after departure), and at the end of the travel (24 h after departure). At each sampling point, 25 fish were sacrificed using an overdose of MS222 (1 g/L Sandoz, Milan, Italy).

### 2.2. Cortisol Measurement

Cortisol was measured in muscle samples obtained from the caudal peduncle of 20 fish, for each sampling time. Muscle samples were frozen by immersion in liquid nitrogen and pulverized. Then, 100 mg of the resulting powder was resuspended in 1 mL of phosphate buffer, pH 7.2 and extracted in diethyl ether. Cortisol was measured with a microtitre RIA (Radioimmunoassay), as described by Bertotto et al. [24]. The procedure was validated for the meagre and the sensitivity of the assay was 3.125 pg well^−1^ [18].

### 2.3. Immunohistochemistry (IHC)

Immunohistochemical staining was evaluated within several organs (skin, gills, stomach, intestine, liver, pancreas, muscle) obtained from 5 fish for each sampling time. Tissues were fixed in 4% paraformaldehyde in phosphate-buffered saline (PBS, 0.1 M, pH 7.4) at 4 °C overnight, rinsed in PBS, dehydrated in a rising alcohol series and embedded in paraffin. Serial sections were cut at 4 µm for the immunohistochemical analysis.

The list of primary antibodies used in this study is consistent with that described in Bortoletti et al. [18] and is reported in Table 1. The Elite ABC KIT system (Vector Laboratories, Inc., Burlingame, CA, USA) was used for IHC, as described in Pascoli et al. [21]. Briefly, blocking of endogenous peroxidase activity was performed by incubating the sections in 3% H_2_O_2_ in PBS. Non-specific binding sites were blocked by incubation with normal goat serum (Dakocytomation, Milano, Italy). Following the incubation in primary antibodies (see Table 1) overnight at 4 °C, sections were washed and incubated with biotin-conjugated anti-mouse Ig antibodies (Dakocytomation, Milano, Italy). The immunoreactive sites were detected using the avidin-biotin peroxidase complex (Vector Laboratories, Inc., Burlingame, CA, USA), adding a solution of 10 mg of 3.3′-diaminobenzidine tetrahydrochloride (DAB, Sigma, Milano, Italy) in 15 mL of 0.5 M Tris buffer at pH 7.6, containing 1.5 mL of 0.03% H_2_O_2_. After IHC, a nuclear counterstaining with Mayer’s haematoxylin was performed. To verify the immunostaining specificity sections were incubated with: (i) PBS instead of the specific primary antibodies; (ii) pre-immune sera instead of the primary antisera; (iii) PBS instead of the secondary antibodies. The results of these controls were negative.

### 2.4. Statistical Analysis

Statistical analysis was carried out by means of STATISTICA 8.0 (StatSoft, Tulsa, OK, USA). After transformation in a logarithmic scale (with a logarithmic function), the cortisol concentrations were compared by means of one-way analysis of variance (ANOVA). In the presence of significant differences between the means, a post hoc comparison of Honestly Significant Difference (HSD) for unbalanced designs was carried out.

## 3. Results

### 3.1. Stress Response

The average cortisol values detected in the muscle of the sampled fish before transport (control), after loading, during transport, and at the end of the travel are presented in Figure 1. A significant increase in cortisol levels was observed when comparing fish from the facility’s holding tanks (2.28 ± 0.47 ng/g) to those in the transport tanks of the truck (6.97 ± 1.37 ng/g; *p* < 0.0001). However, no significant differences in cortisol values were found among the different transport phases (*p* > 0.05), although higher values were noted in fish at the conclusion of the travel (11.16 ± 2.15 ng/g).

### 3.2. HSP70 and Oxidative Stress Markers

Immunopositivity was observed in various tissues and organs for all antibodies, including the skin, stomach, intestine, hepatopancreas, and gills, while the skeletal muscle tested negative. The summary of the immunohistochemical analysis results is presented in Table 2, indicating the positivity and signal intensity levels in various tissues and organs of the animals sampled before and at the end of transportation. The data presented represent the average positivity observed in five animals per experimental group.

Overall, the immunohistochemical analysis revealed the cellular localization of all tested antibodies. However, no significant differences in signal intensity were detected between the control and transported animals, except for HNE intestinal distribution, which was more pronounced in transported animals.

For all antibodies, positive immunostaining was evident in the epithelial cells of the skin, in gill epithelium lining primary and secondary lamellae and in the cells lining the gastric mucosa and gastric glands of the stomach. Additionally, positive staining was found in the enterocytes of the intestinal mucosa as well as in the hepatocytes and pancreatic glands within the hepatic parenchyma.

In all cases, the positive immunostaining was observed at the cytoplasmic level, except for anti-8-OHdG, which exhibited nuclear positivity.

#### 3.2.1. HSP70 Immunohistochemistry

Figure 2 illustrates a series of panels depicting the immunohistochemical localization of HSP70 in various tissues and organs of both control (A, C, E) and transported animals (B, D, F). No significant difference in terms of immunopositivity was observed between the two groups (see also Table 2 for comparison). In the hepatopancreas (Figure 2A,B), immunopositivity was evident in the cytoplasm of hepatocytes comprising the hepatic parenchyma and the acini (glands) of the pancreas. In the stomach (Figure 2C,D), immunopositivity was primarily observed in the gastric glands, while being less pronounced in the pits and epithelium lining the gastric mucosa. Similarly, in the intestine, immunopositivity was noticeable in the cytoplasm of enterocytes within the intestinal mucosa, with positive results also observed in rodlet cells (Figure 2E,F).

#### 3.2.2. HNE, NT and 8-OHdG Immunohistochemistry

Figure 3 illustrates a series of panels depicting the immunohistochemical localization of HNE (A–D), NT (E,F) and 8-OHdG (G,H) in various tissues and organs of both control (A,C,E,G) and transported animals (B,D,F,H). Except for intestinal HNE immunopositivity, no significant differences in terms of immunopositivity were observed between the two groups (see also Table 2 for comparison).

In relation to HNE, positive immunostaining was detected in the cytoplasm of hepatocytes of both control (Figure 3A) and transported animals (Figure 3B), with no discernible differences observed between the two groups. However, in the intestine, a more pronounced positivity was evident in the enterocytes of transported animals (Figure 3C) compared to the controls (Figure 3D), indicating a higher level of lipid peroxidation.

Positive immunostaining to the 3-Nitrotyrosine (NT) antibody was observed in the intestine at the level of the enterocytes (Figure 3E,F).

Figure 3G,H display a stomach positive to the 8-OHdG antibody. The immunoreactivity was observed in the nuclei of the secretory epithelium lining the mucosa, as well as in some of the nuclei of the underlying glands.

## 4. Discussion

The objective of this study was to assess the stress response in meagre juveniles during transportation. The assessment of physiological parameters during stressful operations, such as transportation, yields valuable insights for establishing appropriate management practices [25]. In the present study, we assessed stress parameters, including cortisol levels and the cellular expression of specific oxidative stress indicators.

The muscle cortisol levels in the fish transferred to the transport tanks were significantly higher than those of the controls. Elevations in plasma cortisol levels as a result of transport-induced stress have previously been reported in various marine and freshwater species, such as European seabass (*Dicentrarchus labrax*) [9], Pacific bluefin tuna (*Thunnus orientalis*) [26], Nile tilapia (*Oreochromis niloticus*), channel catfish (*Ictalurus punctatus*) and rainbow trout (*Oncorhynchus mykiss*) [27,28,29,30]. Consistent with other studies, fishing and loading operations onto transport tanks are commonly identified as the most stressful procedures for fish [31,32,33,34]. Before the loading procedure, fish confined in fishing nets tend to exhibit elevated cortisol levels [35]. However, in contrast to our previous study [18], cortisol levels did not show a significant decrease during transportation; instead, they exhibited a non-significant upward trend until the end of the travel. One plausible hypothesis to account for this difference is the variation in transportation duration (48 h vs. 24 h), indicating that the fish transported in the present study might not have had sufficient time for cortisol levels to return to baseline. The limited time available for recovery could be the reason explaining the maintenance of elevated levels in the end-of-travel sampling.

In our prior investigation [18], we observed a noteworthy decrease in muscle cortisol levels after 16 h from departure. However, our current study yielded contrasting results despite employing an equivalent sampling timeframe (16 h) for both transportation methods. Hence, we speculate that additional factors, such as transportation conditions, pre-transportation fish condition, and potential disparities between fish stocks, might have influenced the observed variation. In our analysis, we meticulously assessed essential transportation parameters that influence animal welfare, encompassing water quality and stocking density, and found them to be consistent between both trips.

Another hypothesis is that differences in the fish stocks transported in the two travels led to varied initial conditions. It is possible that some fish within the transported groups may be more susceptible to stressors, such as disturbances caused by inefficient personnel or suboptimal water conditions. It is important to acknowledge individual differences in stress response, with varying degrees of “stress responsiveness” [6]. These individual variations could have contributed to the observed differences in cortisol levels and stress response between the two transportation scenarios.

Furthermore, variations in cortisol levels during the two transports may result from differences in the loading procedure and/or transportation process due to different operators involved. It is possible some fish might have experienced more manipulations or impactful events, such as air exposure, during loading compared to others. The observed differences in transported animals could be influenced by a combination of these factors. This underscores the necessity for precise study, planning, and execution of practices during complex and impactful procedures like transportation. Implementing standardized protocols and minimizing potential stressors can help reduce variability and enhance the accuracy and reliability of experimental results.

In addition to evaluating the primary stress response in the present study, we also investigated the secondary stress response by assessing indicators such as Heat Shock Protein 70 (HSP70) and oxidative stress products (4-Hydroxynonenal (HNE), 3-Nitrotyrosine (NT), and 8-Hydroxy-2′-deoxyguanosine (8-OHdG)) through immunohistochemistry. This technique facilitated the examination of the presence and cellular localization of these molecules of interest using specific antibodies.

The immunohistochemical investigation provided valuable insights into the distribution of different markers across various tissues and organs in both the control and transported animals. Positive results for the various markers were observed in the epidermis of the skin, intestine, gill lamellae, as well as the gastric glands and hepatopancreas. However, no substantial variations in signal intensity were observed when comparing control and transported animals, in accordance with previous studies on the effect of transport stress in marine fish species using immunohistochemical techniques [9,18]. Nevertheless, a notable exception was observed in the case of HNE. In the present study, most tissues exhibited comparable levels of HNE immunopositivity between transported and control animals, except for the intestine, where the signal appeared more pronounced in the transported animals. This observation aligns with the understanding that lipid peroxides, sensitive markers of cellular oxidative stress, can undergo subsequent reactions, leading to the formation of more complex and reactive compounds, including MDA (malondialdehyde) and HNE (4-Hydroxynonenal) [16]. Recent research conducted by Fiocchi et al. [19] provided further evidence of increased HNE immunopositivity in various tissues of European seabass subjected to handling and temperature increase, reinforcing the role of HNE as an indicator of oxidative stress in fish. Additionally, a separate investigation on the effects of transportation on rainbow trout reported a significant increase in MDA content during the transportation process [36], underscoring MDA as a pertinent indicator of oxidative damage [37]. These findings collectively suggest that the transportation process can impact oxidative stress markers in fish, with HNE showing a pronounced response in the intestine. Understanding these oxidative stress indicators contributes to a deeper comprehension of the physiological responses of fish during transportation, thereby enhancing knowledge of fish welfare and health aspects in the context of transportation-related stress.

To summarize, the detected cortisol levels confirmed the presence of a stress response due to transportation, with the pre-transport procedures, particularly the loading process, being a significant stress factor. Following this operation, cortisol levels increased and remained elevated throughout transportation, indicating that the animals did not fully recover from the stress experienced. There are two possible explanations for this: the limited available time or the highly stressful transportation conditions.

With respect to the cellular localization of oxidative stress markers, immunohistochemistry analysis showed, overall, no significant variation before and after transportation, suggesting that the stress experienced by the fish during transportation might have been relatively mild.

Lastly, it is essential to highlight that cortisol continues to be a valuable indicator of well-being in farmed species, and muscle serves as an intriguing matrix for evaluating cortisol levels and monitoring the stress response. Muscle is less influenced by the sampling effect while remaining highly comparable to a blood matrix, making it applicable even in situations where blood sampling is not possible, such as in *post-mortem* examinations.

## 5. Conclusions

This study focused on describing and analyzing the stress response to transportation in meagre juveniles. The observed stress response appeared to be primarily associated with loading stress and the transport process itself, underscoring the significance of implementing appropriate operational protocols to preserve fish welfare. Nevertheless, the overall unaltered distribution of oxidative stress markers between control and transported fish suggests that the experienced stress might have been relatively mild. The use of muscle instead of blood for cortisol evaluation once again proved to be an excellent tool in assessing *post-mortem* stress, particularly when blood sampling is not feasible. These findings contribute to our understanding of the stress factors involved in fish transportation and emphasize the need for further research to enhance fish well-being during transport operations.

## Figures and Tables

**Figure 1 animals-13-03288-f001:**
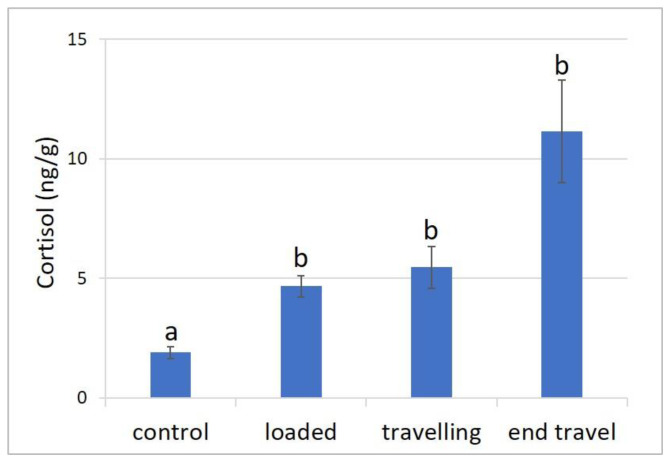
Cortisol concentrations detected in meagre muscle at four phases: before transport (control); after loading; during travel (after 16 h from departure) and at the end of travel (after 24 h). Data are expressed as mean ± SE (*n* = 20). Different letters denote statistically significant differences at the different sampling points (*p* < 0.0001).

**Figure 2 animals-13-03288-f002:**
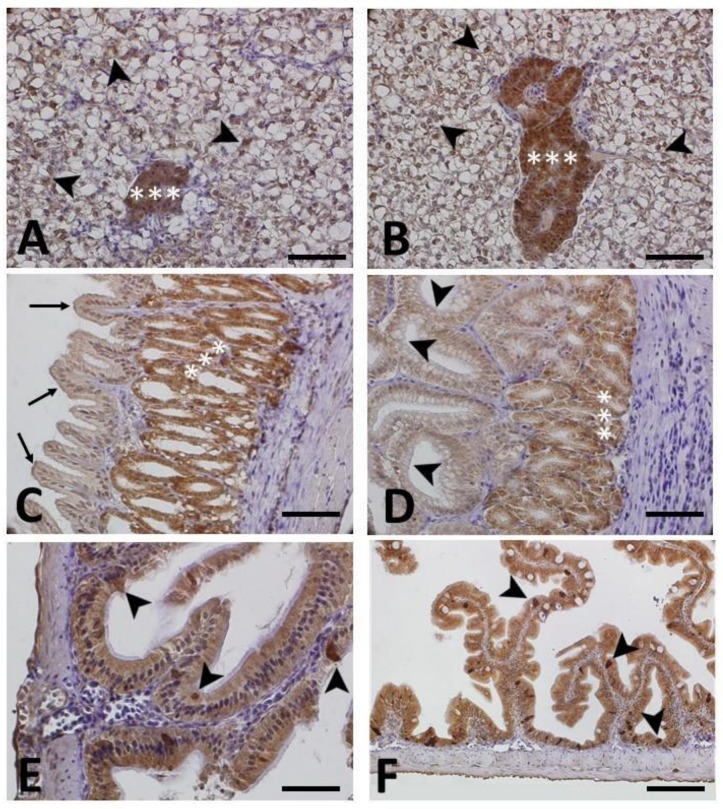
Immunohistochemical localization of Heat Shock Protein 70 (HSP70) in different tissues of control (**A**,**C**,**E**) and transported animals (**B**,**D**,**F**). All sections are counterstained with Mayer’s hematoxylin. No notable variations in immunoreactivity were evident when comparing control and transported animals (refer also to Table 2). (**A**,**B**) Hepatopancreas: marked HSP70 immunostaining is evident in the parenchyma within the cytoplasm of hepatocytes (arrowheads) and in the acini (glands) (asterisks). (**C**,**D**) Stomach: marked immunopositivity is observed in the gastric glands (asterisks), and it is less pronounced in the pits (arrowheads) and epithelium lining the gastric mucosa (arrows). (**E**,**F**) Intestine: marked HSP70 immunoreactivity is detected in the cytoplasm of enterocytes and in the rodlet cells (arrowheads). Scale bar (**A**–**E**): 200 µm; scale bar (**F**): 100 µm.

**Figure 3 animals-13-03288-f003:**
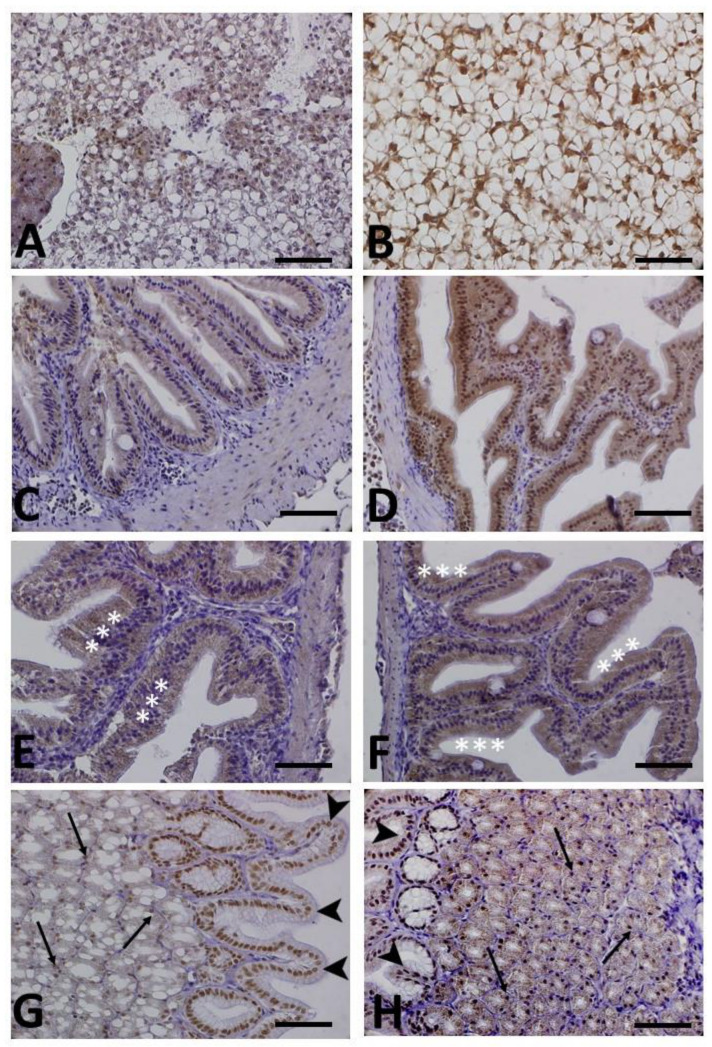
Immunohistochemical localization of 4-Hydroxy-2-nonenal (HNE) in panels (**A**–**D**), 3-Nitrotyrosine (NT) in panels (**E**,**F**) and 8-Hydroxy-2′-deoxyguanosine (8-OHdG) in panel (**G**,**H**) in meagre (*Argyrosomus regius*). Panels (**A**,**C**,**E**,**G**) show control animals, while panels (**B**,**D**,**F**,**H**) depict animals exposed to transport stress. All sections are counterstained with Mayer’s hematoxylin. No notable variations in immunoreactivity were evident when comparing control and transported animals, except for HNE (refer also to Table 2). (**A**,**B**) Hepatopancreas: slight HNE immunostaining is evident in the parenchyma within the cytoplasm of hepatocytes and in the acini (glands). (**C**,**D**) Intestine: moderate (**C**) and marked (**D**) HNE immunopositivity is observed in the enterocytes of control and transported animals, respectively. (**E**,**F**) Intestine: moderate NT immunostaining in the enterocytes (asterisks). (**G**,**H**) Stomach: marked 8-OHdG immunostaining is visible in the nuclei of the secretory epithelium lining the mucosa (arrowheads), as well as in some of the nuclei of the underlying glands (arrows). Scale bar: 200 µm.

**Table 1 animals-13-03288-t001:** Details of the primary antibodies (data from [18]).

Target Protein	Antibody	Species	Dilution	Characterization	Immunizing Antigen/Source
Heat Shock Protein 70 (HSP70)	Anti-HSP70	mouse monoclonal	1:600 (IHC)	Immunohistochemistry	Recombinant fragment from human Hsp70 (Abcam, Cambridge, UK)
4-Hydroxy-2-nonenal (HNE)	Anti-HNE	mouse monoclonal	1:50 (IHC)	Immunohistochemistry	4-Hydroxy-2-nonenal modified KLH (Abcam, UK)
3-Nitrotyrosine (NT)	Anti-NT	mouse monoclonal	1:1000 (IHC)	Immunohistochemistry	3-(4-Hydroxy-3-nitrophenylacetamido) propionic acid conjugated to bovine serum albumin (BSA) (GeneTex, Inc., Irvine, CA, USA)
8-Hydroxy-2′-guanosine (8-OHdG)	Anti-8-OHdG	mouse monoclonal	1:3000 (IHC)	Immunohistochemistry	8-Hydroxy-2’-deoxyguanosine conjugated Keyhole Limpet Hemocyanin (Abcam, UK)

**Table 2 animals-13-03288-t002:** Immunohistochemical detection of Heat Shock Protein 70 (HSP70), 4-hydroxy-2-nonenal (HNE), 3-nitrotyrosine (NT) and 8-hydroxy-2′-deoxyguanosine (8-OHdG) in tissues of meagre (control animals and after transport): -, not detectable; +/-, slight signal; +, moderate signal; ++ marked signal.

Tissue	Control				Transported			
	HSP70	NT	HNE	8-OHdG	HSP70	NT	HNE	8-OHdG
Skin	+	-	-	+	+	-	-	+
Stomach	++	+	++	++	++	+	++	++
Intestine	++	+	+	++	++	+	++	++
Hepatopancreas	++	+/-	++	+	++	+/-	++	+
Gills	++	+/-	++	++	++	+/-	++	++
Muscle	-	-	-	-	-	-	-	-

## Data Availability

The data presented in this study are available within the article.

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
