# Peer review of "Influence of Transportation on Stress Response and Cellular Oxidative Stress Markers in Juvenile Meagre (Argyrosomus regius)"

_animals, 2023, doi:10.3390/ani13203288_

Round 1
Reviewer 1 Report
The paper by Bortoletti and colleague is addressed to evaluate the stress response of the fish Argyrosomus regius to the transportation from the hatchery to the fattening facility. Authors highlight the morpho-physiological effect in terms of welfare. This is a relevant aspect, for welfare and for ensuring high quality of fish meat for consumers. Stressed animals can be prone to develop subclinical symptoms which may create a damage to animals and ultimately cause a decline of the quality of fish meat with an economic burden.
Few concerns:
line 111-112: please revise the Eu Directive currently in place
line 121 on: May author report also the season of transportation? Do they consider that seasonality may affect animal stress response?
line 298 on: among the hypothesis, could the seasonality be a variable influencing the stress response?
Reviewer 2 Report
The authors have investigated the physiological stress response of meagre juveniles during commercial transport. The results show that the transported fish experienced stress within tolerable limits. This research subject undertaken in the study is reasonable. The work could be relevant to the journal’s readership, however, I feel that some of the conclusions presented are supported by the data or within the scope of the study insufficiently. The work needs to be completed. I believe the paper has potential, if the issues specifically covered are adequately addressed and completed.
1. The current study yielded contrasting results compared with the former study showing decrease in muscle cortisol levels. It’s necessary to try another repeat experiment logically. Otherwise, the previously findings are only limited to this very transported groups. The experiments were not well designed. The dynamic metabolic cycle cortisol levels shouldn’t be defined as one way changes. Continual starvation together with transportation makes it more complicated. At least one more sampling experiments 24 hours after the arrival,and one more control group without transportation. Plasma cortisol levels should also be measured as well.
2. All the stress response markers tested as protein level, which might less sensitive in IHC. These are not hormone, or some other cytokine that could be response instantaneously . Changes of RNA expression level might be more sensitive in this experiment, and this might provide more detailed information. The IHC figures are not well presented. There are only some selected images incongruously. It’s better to put the comparable results next to the same tissue tested.
The English is well based on the present study.
Reviewer 3 Report
In the manuscript “Effect of transport on stress response and cellular oxidative 2 stress markers in meagre (Argyrosomus regius) juveniles” the author tackle one of the key issue on fish welfare: the transportation-related stress in a fish species of economic importance. They develop the research to answer the key question through a combined physiological and morphological approach, with rigorous methodology. They use a fish-specific protocols for cortisol measurements as well as they employ antibodies previously tested in fish species.
Minor concerns are:
paragraph 2.1. can they describe the way of fish loading? Is this a variable which might influence the stress response?
figures 2 and 3: to improve the readability, would be good to organize images on two columns (one of control and one related to animals after transportation).
The main question addressed by the research is the level of acute stress related to transportation in a fish species of raising economic importance. The topic is highly relevant dealing with welfare of animals. The importance of fish welfare is growing due to the well established concept that fish are sentient animals. The manuscript adds new insights to the field. The subject area added characterization of antibodies in this fish species.
As further controls, protein expression in the tissues may enhance the morphological staining observations.
